# Intermittent catheter users' symptom identification, description and management of urinary tract infection: a qualitative study

Ikumi Okamoto,[1] Jacqui Prieto,[1] Miriam Avery,[1] Katherine Moore,[2] Mandy Fader,[1] Samantha Sartain,[3] Bridget Clancy[3]

[1]Faculty of Health Sciences, University of Southampton, University Road, Southampton, SO17 1BJ, UK
[2]Faculty of Nursing, University of Alberta, Edmonton, Canada
[3]University of Southampton, University Road, Southampton SO17 1BJUK

**Correspondence to**
Dr Mandy Fader;
m.fader@soton.ac.uk

## ABSTRACT

**Objectives** To elucidate the views of intermittent catheter (IC) users regarding urinary tract infection (UTI) symptom presentation, terminology for describing signs and symptoms, the cause of UTI and management strategies.

**Design** Qualitative study with semi-structured interviews. The transcribed text was analysed thematically.

**Setting** 12 general practitioner (GP) surgeries in Hampshire and Dorset, UK.

**Participants** A convenience sample of 30 IC users, aged over 18, using IC for at least 3 months who had at least one self-reported UTI since starting IC.

**Results** Participants reported a variety of signs and symptoms, such as urine cloudiness and smell, as indicators of UTI. The terms used often differed from those in the modified National Institute on Disability and Rehabilitation Research (NIDRR) symptom set. IC users had difficulty distinguishing possible UTI symptoms from those of their comorbidities. They expressed uncertainty about the cause of UTI, often attributing it to poor hygiene and lifestyle behaviours. Whereas some viewed UTI as an expected consequence of IC use that could be self-managed, others felt more concerned and were more reliant on their GP for support. A range of management strategies was described, including drinking more fluids, increased attention to personal hygiene and self-medicating with antibiotics.

**Conclusions** There is uncertainty among IC users about UTI signs and symptoms and when to seek help. Individual accounts of UTI fitted generally within the modified NIDRR descriptors but adopted less technical and more 'lay' language. IC users' descriptions of UTI signs and symptoms can lack precision, owing partly to the presence of underlying health conditions. This, together with differing levels of concern about the need to seek help and self-medication with antibiotics, presents challenges for the GP. This study provides the basis for developing a self-help tool which may aid identification of UTI and enhance communication with healthcare professionals.

## INTRODUCTION

Intermittent catheterisation (IC) is widely used for the urological management of people with incomplete bladder emptying.[1 2] Urinary

### Strengths and limitations of this study

► This is the first qualitative study to explore intermittent catheter (IC) users' views, experiences and descriptions of urinary tract infection (UTI) signs and symptoms.
► The study includes IC users who had a wide range of reasons for using IC, including both neurological and non-neurological conditions.
► All UTI signs and symptoms reported in this paper were subjective, retrospective and self-reported and were not confirmed by laboratory analysis.
► Advice from healthcare professionals that IC users discussed was not confirmed.

tract infection (UTI) is the most frequently reported and challenging complication.[2] In 1992, medical experts developed a key document on diagnosis and management of UTI in spinal cord injured individuals (SCI) for the National Institute on Disability and Rehabilitation Research (NIDRR).[3] The signs and symptoms listed in 1992 remain the current clinical standard to guide practice for the patients who are neurogenic: positive urine culture plus one or more signs or symptoms (leucocytes in the urine, discomfort or pain over the kidney or bladder or during voiding, onset of urinary incontinence, fever, increased spasticity, autonomic dysreflexia, cloudy malodorous urine or malaise, lethargy, sense of unease). However, the document did not include patient contributions from non-SCI IC users to the terminology or to the presenting symptoms and therefore cannot be considered to fully reflect the experience of the wider IC population. As UTI signs and symptoms can be difficult for some IC users to interpret, the NIDRR descriptors may be perceived differently or may not directly apply[2 4] and thus can be difficult for an individual to describe to their

general practitioner (GP). Massa *et al*[5] used a UTI signs and symptoms list modified from the NIDRR criteria[3] with SCI individuals using IC and found that only 66% could self-predict their own UTI. Research exploring how IC users themselves identify and describe the signs and symptoms of UTI, as compared with the existing criteria, may aid in understanding how these criteria are interpreted and when assistance is sought. Patient-oriented language may also assist in mutual understanding of the presence of a UTI.

Although there are studies exploring experiences and issues related to IC,[6–12] UTI has not been a focus. We found no qualitative research specifically exploring IC users' descriptions of UTI signs and symptoms or how these descriptions compare with the existing symptom list. Thus, in this study, we conducted qualitative interviews with IC users with the aim of investigating their descriptions of UTI signs and symptoms, including their strategies for prevention, identification and management, with the goal of informing further research on a symptom-specific tool for IC users.

## METHOD
The study used a qualitative method. Convenience sampling was used to recruit participants using IC on a regular basis from 12 GP practices in Hampshire and Dorset, UK to take part in interviews regarding their experiences of IC as part of a larger study (the MultiCath programme[13]). Participants were over 18 years of age, had independently used IC for at least 3 months and had no reported urethral deformities or immune deficiency disease. Ethical approval was obtained from the appropriate local research ethics committees. Written consent was obtained from all participants.

### Data collection
One face-to-face interview lasting 45–60 min was conducted in each participant's home by one of three experienced female interviewers including a qualitative researcher (SS) and research nurses (BC and MA). None of the interviewers had had previous contact with any of the participants, and they attended as a non-clinical and non-judgemental listener. If desired, a relative or carer of the participant was present. A semi-structured interview schedule (see online supplementary appendix 1) was used to explore experiences, values, beliefs and themes around the use of catheters generally. Based on UTI-related symptoms identified from the first six interviews, we then added haematuria to the modified NIDRR symptom list[5] (see online supplementary appendix 2). The list was then used in subsequent interviews. Participants were prompted to describe their experience of UTI signs and symptoms in their own words. They were also shown the symptom list to go through with the interviewer, which helped participants recall further signs and symptoms they had experienced. Recruitment continued until data saturation was reached. Interviews were recorded and then transcribed verbatim. All transcripts were anonymised with an assigned study number. The transcripts were read only by the researchers and not by participants. Field notes were made during the interviews, which were used in the analysis.

### Data analysis
A qualitative researcher (IO), who had not conducted the interviews, analysed the data thematically and coded it using NVivo 10 (QSR International). The first 20 transcripts formed an initial coding framework. Coding was further refined through discussion with the wider research team and ongoing analysis of the remaining transcripts. Newly emerged themes that did not fit into the initial themes were discussed as a team. No participants were involved in the analysis of data.

## RESULTS
### Participants
One hundred and thirty-nine IC users were invited to take part and 74 (53%) responded, of whom 42 were willing to participate. Three were ineligible. Nine participants had never experienced a UTI since starting IC and were not asked any further questions regarding UTI experiences during the interview. Therefore, the findings presented here are based on a subset of our sample. The total number of participants was 30, including 19 men and 11 women, with mean age 66 years (range 23–86), using IC for approximately 10 years (range 9 months–31 years). The frequency of IC ranged from 1 to 10 catheterisations a day (mean 4/day). Reasons for IC included chronic urinary retention (n=15), neurological impairment (n=10) and other factors such as postsurgery or postchemotherapy (n=5). Three were currently re-using their catheters but as they were few in number, no distinction was made between re-users and single-users during data analysis. All participants were English speakers. No race/ethnicity data were collected.

### Findings
Three themes were identified regarding experiences of UTI: (1) IC users' description of their signs and symptoms and degree of certainty; (2) help-seeking actions when UTI is suspected and (3) understanding of causation and prevention strategies used.

#### IC users' description of their signs and symptoms and degree of certainty
##### Terms used to describe signs and symptoms
The participants reported a variety of signs and symptoms that they thought were caused by UTI. Table 1 provides examples of participants' descriptions of the UTI signs and symptoms collected during the interviews with a corresponding comparison to the modified NIDRR criteria.[5] Participants were likely to use informal terms to describe their signs and symptoms such as 'cannot hold my water' or 'having to go more frequently' rather than

**Table 1** Participants' descriptions of UTI signs and symptoms and mapped to modified NIDRR symptom list

| Symptom categories | Examples of participants' descriptions of UTI signs and symptoms collected during the interviews | Mapping to the modified NIDRR symptom list[5] |
|---|---|---|
| Changes to urine | ▶ Different colour<br>▶ Cloudiness<br>▶ Darkness in the urine<br>▶ A bit like rather thick soup | Cloudy urine |
| | ▶ The smell of urine not being right<br>▶ A really strong smell<br>▶ Smell<br>▶ Horrible smell<br>▶ Change in the odour of the urine | Foul smelling urine |
| | ▶ Looked like there was a little bit of blood present<br>▶ Specks of blood | Haematuria* |
| | ▶ There was protein present (found at general practice) | Leukocytes in the urine |
| Changes in bladder emptying | ▶ Feel like you need to go more often but not passing enough<br>▶ I was going to the toilet, and then not feeling as I'd actually been; there was still a pressure there of wanting to go<br>▶ You feel you want to keep having a pee constantly<br>▶ Having to go more frequently<br>▶ Can't pass water properly<br>▶ Be able to have a normal pee without having to use a catheter<br>▶ I feel as though I empty my bladder continually<br>▶ I've got to do (intermittent catheter) twice as much<br>▶ Going to toilet less | Increased frequency of catheterisation |
| | ▶ Leak a bit between catheterisations<br>▶ Cannot hold my water; a couple of accidents during the day | Incontinence |
| Fever | ▶ Hot and sticky<br>▶ Bad fever, high temperature<br>▶ Very hot and shivery, not much of a temperature<br>▶ A fever symptom | Fever |
| Generalised/systemic symptoms | ▶ Not feeling well<br>▶ I feel low, very rundown<br>▶ Don't feel right<br>▶ Lesser feeling of well-being; slight feeling of discomfort, not quite right<br>▶ Don't feel comfortable | Feeling tired |
| | ▶ I feel like I don't want to go anywhere and I feel tired<br>▶ Don't feel right in yourself | Sense of unease |
| | ▶ Vertigo<br>▶ Felt as if I was going to pass out<br>▶ Couldn't eat anything; constantly being sick; diarrhoea | Feeling sick |
| Pain/discomfort | Discomfort or pain over bladder<br>▶ The bladder gets very irritated; the whole pelvis is quite irritated<br>▶ Burning bladder<br>▶ The pain usually starts in the urethra area<br>▶ Slight irritation (in my urethra and around that area)<br>Discomfort or pain over kidney<br>▶ Tremendous pain in my back, in my kidney area<br>▶ Dull ache in my lower back and stinging | Discomfort or pain over the kidney or bladder |

Continued

**Table 1** Continued

| Symptom categories | Examples of participants' descriptions of UTI signs and symptoms collected during the interviews | Mapping to the modified NIDRR symptom list[5] |
|---|---|---|
| | Pain during urination<br>► Get a pain when you are having a pee<br>► Burning when I pass urine<br>► Very difficult to pass water<br>► When I go to the loo, it slightly burns<br><br>Discomfort or pain in penis/groin area<br>► Stinging feeling in your penis itself<br><br>Non-localised discomfort/pain<br>► Feel uncomfortable down below<br>► Get uncomfortable; when I sit down, feels like there is something there; uncomfortable to walk<br>► It's really really painful<br>► I wouldn't say painful, but it's discomfort<br>► Pain, stinging sometimes<br>► Feeling uncomfortable; even if I've used a catheter, still feels uncomfortable<br>► It burns like mad. There's no part that doesn't burn<br>► A tenderness all around the area<br>► Everything down there feels very irritable; feel very uncomfortable<br>► It gets extremely painful | |
| Not reported | Autonomic dysreflexia | |
| Not reported | Increased spasticity | |

*An item added to the modified NIDRR list for our symptom list used during the interviews.
NIDRR, National Institute on Disability and Rehabilitation Research; UTI, urinary tract infection.

'incontinence' and 'increased catheterising', respectively (for more examples, see table 1). If they described pain, it was often generalised and not localised until they were prompted. Terms such as 'discomfort' and 'uncomfortable' were used to describe both general physical symptoms and pain or even increased urinary frequency—'*everything feels very irritable … in the area of the penis or the bladder. You just feel very uncomfortable …as if you want to go all the time*' (Participant 41, male). Corresponding terms in the modified NIDRR set such as 'discomfort or pain over the kidney or bladder' did not cover the breadth of symptoms experienced. The language used to describe the signs and symptoms often differed from typical clinically accepted terms; however, none of the participants voiced concern about being unable to communicate effectively with GPs.

### Uncertainty about whether symptoms were caused by UTI

Deciding if they had a UTI was difficult for some participants because of comorbid conditions. For example, one with bladder cancer and 'prostate problems' sometimes had a feeling of irritation but was unsure whether symptoms were caused by UTI or by his other health problems:

*[Talking about UTI is] difficult because I've always had problems down below. I mean I [have] prostate problems,*

*bladder problems, bladder cancer, a lot. So sometimes, I get this irritation feeling but [it] never bothers me. (Participant 12, male)*

Others also mentioned the difficulty of distinguishing UTI symptoms from their age-related problems.

*I'm eighty-four and it's one of those things, you expect things to happen as you're getting old. Aches and pains, and stuff like that, it could be to do with that, I don't know, but you get so many bits and pieces. (Participant 4, male)*

### Help-seeking actions when UTI is suspected
#### Decision process whether to seek help or not

Many participants reported that cloudy or smelly urine were the first indicators; however, how they managed such signs varied considerably. Most usually waited a few days to be convinced that their symptoms were caused by UTI before contacting their GP and used 'self-help' strategies such as increasing fluids:

*[I]try to drink plenty and sort of flush it through and obviously if after a day or two I still felt bad or got worse I would have gone to the doctor's. (Participant 3, female)*

Others waited until the onset of other symptoms, such as pain and discomfort, in addition to the first signs.

*I think it's a little bit cloudy, but the next day it's perfectly clear. So I think one gets this in the way, sometimes, and if it is notably cloudy and I'm getting the pain [in the penis], it becomes pretty obvious that actually that's what I've got. (Participant 32, male)*

Another stated:

*Since [I started IC] I have had and may well even now have a urinary infection, but if I drink enough it just doesn't affect me. Would I want to be on antibiotics all the time? […] I think it is more important that [UTI] is not affecting your overall health. I probably have an infection at the moment but it doesn't affect me in any way, I don't have a temperature, I'm not ill as a result of it. I actually had it out with my doctor a bit because they did a test at one point and said that I had an infection and I must take antibiotics. I thought I can but I'll probably have another one in 3 or 6 months time. (Participant 37, male)*

Conversely, others contacted the GP at the first sign, such as cloudy/smelly urine, as previous experience suggested that more severe symptoms would follow if left untreated:

*I always know as soon as I've got [an infection] because as soon as it's there, the smell's there. And if I don't sort of act on it straight away … within a week I'll be feeling really unwell. […] And if I leave it later, it takes a lot longer to get rid of it. (Participant 11, female)*

A participant's decision to consult their GP was also influenced by previous advice they had received about UTI. For example, a participant with cauda equina who '*lost most feeling below the waist*' said his GP had told him,

*'You probably will know when you've got [UTI] if you get this sort of sickly-sweet smell.' (Participant 10, male)*

### Medication with antibiotics

If self-help strategies weren't effective, many contacted the GP for a prescription for antibiotics. There were three participants who self-medicated with antibiotics, which they kept at home. One felt that:

*If you keep on having antibiotics that eventually it's not going to work. (Participant 13, male)*

For another, a supply of antibiotics was important as it reassured him that he had ready access to treatment:

*It's funny, but if I had an infection, it's usually on a weekend, I can't catch anybody to get the antibiotics or whatever I need, so then I'd have to go to the hospital and all that palaver. So I just keep one in stock all the time. (Participant 19, male)*

*[…] Sometimes (infection) will start by a burning in my willy [penis], and then it will back up further to the bladder, (but) I don't have that problems anymore, because I immediately put a pill into my mouth and off I go and 3 days later it's gone. (Participant 19, male)*

Seven took daily prophylactic antibiotics. One, using IC for 3 years, said:

*I still get infections, off and on. You know when you get an infection, but I've taken tablets all the time for it. Every day, for infection. The doctor gave them to me, saying 'as soon as you're on a catheter, you've got to take these.' (Participant 4, male)*

One participant mentioned that she used a urine dipstick test along with self administered antibiotics as suggested by the GP. She valued this approach and explained that it enabled her to act on a UTI promptly and avoid a trip to the GP surgery and risk seeing a physician who did not know her history:

*You've got all the signs for a water infection [smell of urine, feeling hot and sticky, and leaking of urine], by the time you've got that appointment, you could have had that water infection for 5 days so you know what my old GP did? He sort of said you can go on the internet and you can buy the sticks to test your water yourself which we did, so I've got the sticks myself and my GP once a year tends to give me a prescription a repeat prescription for a couple of courses of antibiotics. (Participant 6, female)*

### Relationship with GPs

Having a good relationship with and easy access to the GP was greatly valued:

*[My GP]'s obviously well aware of where I'm at and he looks after me very well. I'm quite happy to consult with him on what I'm doing. I think three, four, I don't know maybe even five occasions [in the last 6 years] I've had urinary infection [smell and lesser feeling of well-being], he gives me antibiotics straight away and they disappear in no time at all. (Participant 10, male)*

Another emphasised the importance of seeing the same GP who '*understands what I've been going through*' and '*knows my history*':

*…he understands, he knows what I've been through, when you have to explain it to somebody else, it's all there in the notes, whether they look back in anything, but you have to keep explaining things, whereas Dr [name]) understands what I've been going through. (Participant 23, female)*

### Understanding of causation and prevention strategies used
### Clean/correct technique

Many attributed UTI to faulty IC technique—for example, accidentally touching the catheter or not washing hands well enough, but were not sure what they could do differently to prevent a UTI:

*I've got [infections no more than twice a year], obviously didn't do it well enough. It's not good enough really, I mean soap is good and is good, perhaps I should leave it longer, perhaps the [name of disinfectant] should have longer to work, I don't know, I really don't know, I don't know what the time is for [name of disinfectant].(Participant 28, male)*

Public toilets were seen as a potential cause, because such environments are perceived as a source of infection and it was more difficult to carry out IC correctly. For one man, his decision to avoid public toilets limited his outings:

*The problem was when I first started catheterising and going out, I was still getting infections from using public toilets things like that, so I decided I'd change the way I did things and I only go out for half a day now. So I come back midday. So I try not to catheterise when I'm out. (Participant 40, male)*

### Emptying bladder completely

Maintaining a regular schedule of bladder emptying was considered an important preventative strategy. One participant noted:

*In the early stages [I had infections] every couple of weeks and it was a bit worrying at first. […] Sometimes I wouldn't [catheterise] twice a day, I'd only do it once. I realised that I've got to do it twice a day. […] I think making sure I do it twice a day and making sure everything is sterile. (Participant 1, male)*

### Good diet and lifestyle

Good diet and lifestyle was another prevention strategy— no alcohol, caffeine or sugar plus walking were seen as healthy habits to help prevent UTI:

*To try and prevent infection, I don't drink alcohol. Not for 3 years. I don't have caffeine. I was concerned that I was taking chocolate as an energy boost but I am really limiting anything with it; I don't have sugar in anything, I've not had sugar in anything for about 35 years. But I was having the odd biscuit and chocolate and the intake was increasing and I was thinking 'hold on, this has got to stop' so I try and do things that will prevent infection anyway. I drink quite a lot of water. […] I have a walk about 20 min each day. (Participant 41, male)*

## DISCUSSION

The study shows that there is uncertainty among IC users about signs and symptoms of UTI and when to seek help; furthermore, individual descriptions of UTI signs and symptoms often fit only generally within the modified NIDRR descriptors and may require more user friendly language. The non-specific nature of many UTI signs and symptoms presents a challenge for the GP, who relies on the IC user's description, together with urine culture results, to determine the most appropriate course of action.[2] Cloudy or smelly urine was often cited as the first indicator and while some acted on this alone and visited the GP, others described how they would first attempt to self-manage and monitor for further symptoms, such as localised pain, discomfort or generalised illness. Some

with comorbidities and/or age-related problems had difficulty separating a possible UTI from other more general signs and symptoms, which means that there could be the risk of delayed diagnosis of UTI.

The language used to describe the signs and symptoms may differ from typical clinically accepted terms. This is an important finding, as such discrepancy affects the diagnosis and management of UTI among IC users. No participants, however, expressed concern about being unable to communicate effectively with GPs. This may be due to their ongoing relationships with their GP so that language or interpretation differences may have been less relevant. Comprehension of terminology surrounding IC could be a focus for further research and help equip new IC users with the necessary confidence to advocate for themselves or to manage symptoms. The decision to seek medical assistance was influenced by (1) an individual's perception of symptom severity, (2) the level of confidence in the effectiveness of self-help strategies and (3) the previous experience of UTI and its management by the GP.

The key role of the GP in helping the individual manage, whether with symptom-based treatment or having a repeat prescription, was seen by participants as invaluable. Although some were uncertain about the symptoms of UTI, in general, people using IC on a long-term basis, become very familiar with the warning signs of health issues and want to be proactive in their care. In this study, IC users indicated management was facilitated by the positive relationship and continuity with their GP who was aware of their health status and approached care on an individual basis. This interdependence of the individual and the physician is crucial and the reliance alone on laboratory testing should not define treatment in this population.[3] That self-medication with antibiotics might help IC users feel more confident and secure about managing symptoms needs further exploration.

Finally, healthcare professionals need to be attuned to individuals who self-blame ('not hygienic enough') or lack an understanding of the aetiology of UTI. Such feelings were highlighted in at least one other study involving women (non-IC users) where they attributed UTI to 'poor hygiene', 'negligence', 'not drinking enough' or a 'penalty of growing old'.[14] In our study, some participants unnecessarily restricted their lives by following self-imposed rigid and complex procedures (ie, not going out all day so they could catheterise only at home; restricting their diets; never using public toilets). When an IC user approaches the GP for assistance, exploring the story and probing for the effect of IC on daily life is likely to help in making a collaborative plan for evidence-based self-care. This requires the IC user to understand signs and symptoms which are relevant for them and to have the language to articulate their concerns to the GP. Many of the individuals in our study had used IC for several years

and might have benefited from an updated educational session on current best practice with IC, causes of UTI and use of antibiotics.

## Strengths and limitations

This is the first qualitative study to explore views and experiences of UTI from a focused IC user perspective. A strength is the inclusion of participants with a variety of reasons for using IC (including both neurological and non-neurological conditions). While we included adults of all ages, the majority of our sample consisted of older people (mean age 66 years old) and, thus, does not represent all IC users. In common with other qualitative research, the sample size was small and collecting data through interviews relies on recall of experiences. All descriptions of UTI discussed were subjective and self-reported. Diagnoses were not confirmed so that described signs and symptoms may not have been caused by UTI. Participants talked generally about their antibiotic use for UTI treatment; however, the interview schedule included no questions specifically related to attitudes to or understanding of antibiotics or the evidence related to UTI and IC. Advice on issues, such as use of urine dipsticks, may have been a misrepresentation of what the individual was actually told.

## Implications for future research or clinical practice

To address the most appropriate course of action regarding UTI in IC users, there is a need to develop a more user-based UTI symptom list as well as an evidence-based algorithm for self-care and help seeking. Such a list may also assist in self-management and appropriate antibiotic use in this population. Our findings showed that UTI signs and symptoms could be grouped under five broad categories: (1) changes to urine, (2) changes in bladder emptying, (3) fever, (4) generalised/systemic symptoms and (5) pain/discomfort. Questions focused on these categories may be helpful for both patients and healthcare professionals in elucidating signs and symptoms of UTI. Further exploration of the subjective descriptions of UTI signs and symptoms and matching these with laboratory confirmation is needed in order to aid both the IC user and the healthcare professionals in providing best care.

## CONCLUSION

The study has provided insight into how participants construct their experience of UTI and identified some opportunities for further research particularly related to ongoing IC user education and follow-up, antibiotic use and self-care practices. Current guidelines[15] state that cloudy or malodorous urine in the catheterised adult (including IC users) should not be used alone to differentiate asymptomatic bacteriuria from infection or as an indication for urine culture or antimicrobial

therapy, yet in the current study these were commonly reported changes by IC users and were sometimes the impetus for help-seeking.

Focused follow-up according to evidence-based guidelines may assist in self-management and appropriate antibiotic use. Examining GP's perspectives would provide a balanced approach to improve our understanding of the diagnosis and management of UTI among IC users. This study provides the basis for developing a self-help tool which may aid identification of UTI and communication with healthcare professionals.

**Acknowledgements** We would like to thank all the interviewees who agreed to take part and share their experiences. We thank also Samantha Sartain, PhD and Bridget Clancy, RGN, BSc for conducting part of the data collection and Margaret Macaulay, RGN, MSc for assistance with the manuscript.

**Contributors** MF was the principal investigator and was primarily responsible for the original grant application. MF, JP, MA and SS were involved in designing the study and developing the methods. JP led the overall study team. SS collected the majority of the data. BC and MRA also contributed to the data collection. IO led the analysis and interpretation of the qualitative data and wrote drafts of the manuscripts. IO, JP, MRA, KNM and MF wrote drafts and critically revised the manuscripts.

**Funding** This paper refers to independent research funded by the National Institute for Health Research (NIHR) under its Programme Grants for Applied Research (PGfAR) (Grant Reference Number RP-PG-0610-10078).

**Disclaimer** The views expressed are those of the authors and not necessarily those of the NHS, the NIHR or the Department of Health.

**Competing interests** None declared.

**Patient consent** Detail has been removed from this case description/these case descriptions to ensure anonymity. The editors and reviewers have seen the detailed information available and are satisfied that the information backs up the case the authors are making.

**Ethics approval** NHS Research Ethics Committee (NRES committee London, Hampstead).

**Provenance and peer review** Not commissioned; externally peer reviewed.

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
