## [Reviewer comments · BMJ Open]

ARTICLE DETAILS

TITLE (PROVISIONAL)	Intermittent catheter users' symptom identification, description, and management of urinary tract infection: a qualitative study
AUTHORS	Okamoto, Ikumi; Prieto, Jacqui; Avery, Miriam; Moore, Katherine; Fader, Mandy

VERSION 1 - REVIEW

REVIEWER	Raphaële GIRARD Hospices Civils de Lyon France
REVIEW RETURNED	02-Mar-2017

GENERAL COMMENTS	Thank you for the opportunity to read your text. This work could be useful for the development of better tools for health care workers in relation with patients with IC, but should be take into account for prevention program too. Review of BMJ Open 2017 016453 This paper is very interesting, especially for the health workers in contact with patients using intermittent catheter. This using is clearly more frequent now and many general practitioner don't have pertinent knowledge. The subject is clearly explained, the proposition for revision of the NIDRR tool is pertinent and more adapted. I have any remarks: I have no found the ethics data? The study is based on interviews but the text gives no data concerning ethic committee or written consent form. A description of the pathology of patients, the duration of IC using, and the devices used seems useful. The major signs could be very different between paraplegic patients or post-surgery urological patients, for example. The perception of pain or of emptying perception is different, so the limited % of patient describing this signs could be modified. The perception of correct clean environment could be modified by the availability of single use sterile lubricated catheter. Are these
--

	devices available for all patients of this study? Concerning the form of the paper, the text is a little too long. Perhaps it is possible to limit it. In the Abstract, NIDRR is not defined.
--	--

REVIEWER	Gareth Parry Institute for Healthcare Improvement USA
REVIEW RETURNED	24-Apr-2017

GENERAL COMMENTS	Intermittent catheter users' symptom identification, description, and management of urinary tract infection: a qualitative study This is an interesting paper, that seeks to describe the experiences of intermittent catheter users on their identification of and management of urinary tract infections. The authors describe some important and interesting findings related to how users communicate their experiences, relative to the more technical language used in health care. Abstract: 1) The abstract represent a good summary of the paper. The authors should spell out NIDRR in full in the abstract. Introduction 2) The introduction does a good job of providing the background to the study, in particular, the lack of patient or IC user perspectives when identifying a UTI. Moreover the authors make a strong case for using a qualitative approach. Methods 3) The authors start the methods section with a description of how they intended to sample patients. This seems appropriate. However given the age range 18+ contains a wide range with the potential for a wide range of age-related variation in IC and UTI experience, can the authors add any considerations they gave to sampling specific age groups. In addition, can the authors provide any information on sampling by race/ethnicity or use of English language was considered? 4) Under "Data Collection", although the authors refer to the full semi-structured interview guide being available in Appendix 1, I think it would help a reader for the authors to summarize their overall approach to the interviews sooner – for example if they were asking about the participants experiences in some broad areas, describe those areas a little. 5) The "Data Analysis" section is clear and follows frequently used qualitative approaches. Results 6) The "Participants" section provides a clear description of the participants. Further to my comment 3) above, did the authors have any information on the race/ethnicity of patients? 7) Under the section "Experiences of UTI and description of their signs and symptoms", the results seem to drift into a quantitative,
--

	rather than qualitative form of presentation. From a qualitative perspective, I was not sure what to make of Table 1. Can the authors explain where the words in the column labelled “Perceived signs and symptoms of UTI described by IC users” came from? Can the authors explain more what, given the sample is purposive and not representative, should a reader take from the column labelled “N (%)”? Also, can they provide a summary of what they expect the reader to take from Table 1? 8) The section “Terms used to describe signs and symptoms” makes an important point that interview respondents tended to use different terminology from that of the NIDRR. This seems important, and I think it would be worth the authors expanding a little on whether they think this is important, and how so, in this section. 9) Similar to 8) above, I think the following sections would benefit from a little more interpretation within each section. Discussion 10) The discussion is well written, clear and focused on the results. The authors describe several important issues related to patient’s understanding of IC and UTIs and their relationship with a GP. 11) The section titled “Implications for future research or clinical practice” gets a little speculative in including a Table 2, which seems to suggest what needs to happen. I don’t think the authors have sufficient evidence from this study to propose something that specific yet. However, they make a strong case for future research focusing in this area.
--	---

VERSION 1 – AUTHOR RESPONSE

Reviewer 1: Dr Raphaële GIRARD

This paper is very interesting, especially for the health workers in contact with patients using intermittent catheter. This using is clearly more frequent now and many general practitioner don’t have pertinent knowledge. The subject is clearly explained, the proposition for revision of the NIDRR tool is pertinent and more adapted.

Thank you for your comments and suggestions, which we found very useful as we approached our revision. Please find our responses in the following sections.

I have no found the ethics data? The study is based on interviews but the text gives no data concerning ethic committee or written consent form.

In the ‘Method’ section, the following has been inserted; ‘Ethical approval was obtained from the appropriate local research ethics committees. Written consent was obtained from all participants’.

More detailed information about the ethics is found in the last page after ‘Ethical approval’.

A description of the pathology of patients, the duration of IC using, and the devices used seems useful.

These factors vary very much between the participants. We believe that the current descriptions of the participants under ‘Participants’ sufficiently summarise all these factors.

The major signs could be very different between paraplegic patients or post-surgery urological patients, for example. The perception of pain or of emptying perception is different, so the limited % of patient describing this signs could be modified.

Yes, we agree. %s and the numbers of participants have been now removed from Table 1. (Please see our reply to the reviewer 2’s comment 7 for further explanation)

The perception of correct clean environment could be modified by the availability of single use sterile lubricated catheter. Are these devices available for all patients of this study?

Yes, all participants had sterile single-use catheters which were available on prescription from their

GPs.

Concerning the form of the paper, the text is a little too long. Perhaps it is possible to limit it.

We have reduced the word count slightly.

In the Abstract, NIDRR is not defined.

NIDRR is now fully spelled out in the abstract.

Reviewer 2: Dr Gareth Parry

This is an interesting paper that seeks to describe the experiences of intermittent catheter users on their identification of and management of urinary tract infections. The authors describe some important and interesting findings related to how users communicate their experiences, relative to the more technical language used in health care.

Thank you very much for your thorough review of our paper. We are grateful for the time and energy you expended on our behalf.

Abstract:

1) The abstract represent a good summary of the paper. The authors should spell out NIDRR in full in the abstract.

NIDRR is now fully spelled out in the abstract.

Introduction

2) The introduction does a good job of providing the background to the study, in particular, the lack of patient or IC user perspectives when identifying a UTI. Moreover the authors make a strong case for using a qualitative approach.

Thank you.

Methods

3) The authors start the methods section with a description of how they intended to sample patients. This seems appropriate. However given the age range 18+ contains a wide range with the potential for a wide range of age-related variation in IC and UTI experience, can the authors add any considerations they gave to sampling specific age groups. In addition, can the authors provide any information on sampling by race/ethnicity or use of English language was considered?

Yes, there was the potential for a wide range of age-related and IC usage length-related variation in UTI experience. In 'Strengths and limitations', I have added the following; 'Whilst we included adults of all ages, the majority of our sample consisted of older people (mean age 66 years old), and thus, does not represent all IC users.'

Regarding information on race/ethnicity and use of English of the participants, I have added the following information in 'Participants' under 'Results'; 'All participants were English speakers. No race/ethnicity data were collected'.

4) Under "Data Collection", although the authors refer to the full semi-structured interview guide being available in Appendix 1, I think it would help a reader for the authors to summarize their overall approach to the interviews sooner – for example if they were asking about the participants experiences in some broad areas, describe those areas a little.

More information on the semi-structured interview guide has been added in 'Data collection' under 'Methods'. Please see below.

'A semi-structured interview schedule (Appendix 1) was used to explore experiences, values, beliefs and themes around the use of catheters generally.'

5) The "Data Analysis" section is clear and follows frequently used qualitative approaches.

Thank you.

Results

6) The "Participants" section provides a clear description of the participants. Further to my comment 3) above, did the authors have any information on the race/ethnicity of patients?

As above (see comment 3).

7) Under the section "Experiences of UTI and description of their signs and symptoms", the results seem to drift into a quantitative, rather than qualitative form of presentation. From a qualitative perspective, I was not sure what to make of Table 1. Can the authors explain where the words in the

column labelled “Perceived signs and symptoms of UTI described by IC users” came from? Can the authors explain more what, given the sample is purposive and not representative, should a reader take from the column labelled “N (%)”? Also, can they provide a summary of what they expect the reader to take from Table 1?

“Perceived signs and symptoms of UTI described by IC users” has been replaced with “Examples of participants’ descriptions of UTI signs and symptoms collected during the interview” in Table 1. We agree with you. Given that the number of participants is small and not representative of all IC users, our data are difficult to quantify. The column labelled “N (%)” has been removed from Table 1. Similarly, the whole text under ‘1.1 Signs and symptoms of perceived UTI’, which described the quantitative data of Table 1 was removed.

A brief summary of the table has now been provided under ‘1.1 Terms used describe signs and symptoms’.

8) The section “Terms used to describe signs and symptoms” makes an important point that interview respondents tended to use different terminology from that of the NIDRR. This seems important, and I think it would be worth the authors expanding a little on whether they think this is important, and how so, in this section.

The text under ‘1.1 Terms used to describe signs and symptoms’ has been expanded, and we now discuss this theme more in Discussion by adding the below.

‘The language used to describe the signs and symptoms may differ from typical clinically accepted terms. This is an important finding, as such discrepancy affects the diagnosis and management of UTI amongst IC users. No participants, however, expressed concern about being unable to communicate effectively with GPs. This may be due to their on-going relationships with their GP so that language or interpretation differences may have been less relevant.’

9) Similar to 8) above, I think the following sections would benefit from a little more interpretation within each section.

‘1.2 Uncertainty about whether symptoms were caused by UTI’ is now more discussed in Discussion. For instance, the following has been now inserted.

‘Some with co-morbidities and/or age-related problems had difficulty separating a possible UTI from other more general signs and symptoms, which means that there could be the risk of delayed diagnosis of UTI.’

Regarding the other themes, I believe that the interpretations currently included in Discussion are sufficient.

Discussion

10) The discussion is well written, clear and focused on the results. The authors describe several important issues related to patient’s understanding of IC and UTIs and their relationship with a GP. Thank you.

11) The section titled “Implications for future research or clinical practice” gets a little speculative in including a Table 2, which seems to suggest what needs to happen. I don’t think the authors have sufficient evidence from this study to propose something that specific yet. However, they make a strong case for future research focusing in this area.

I agree. Table 2 has been removed, and we have inserted some sentences which suggest the possible use of questions about UTI derived from our findings in future studies. Please see below. ‘Our findings showed that UTI signs and symptoms could be grouped under 5 broad categories: 1) changes to urine; 2) changes in bladder emptying; 3) fever; 4) generalised/systemic symptoms; and 5) pain/discomfort. Questions focused on these categories may be helpful for both patients and healthcare professionals in elucidating signs and symptoms of UTI.’

Besides all the amendments that we have mentioned above, we have also made some changes to the manuscript, which are all shown in the attached version using track changes.

Regarding Table 1, not only the 4th column showing N and % was removed as suggested by the reviewers, we have also made a change to the third column. The participants’ descriptions of UTI signs and symptoms are now mapped to the modified NIDRR symptom list (in Massa et al) rather

than the NIDRR symptom list. This is because we thought it would be more reasonable to compare the participants' descriptions to the list of symptoms we used during the interviews (which was constructed using Massa's modified NIDRR symptom list) rather than the NIDRR list.